# SOURCE-FREE DOMAIN ADAPTATION VIA DISTRIBUTIONAL ALIGNMENT BY MATCHING BATCH NORMALIZATION STATISTICS

## ABSTRACT

In this paper, we propose a novel domain adaptation method for the source-free setting. In this setting, we cannot access source data during adaptation, while unlabeled target data and a model pretrained with source data are given. Due to lack of source data, we cannot directly match the data distributions between domains unlike typical domain adaptation algorithms. To cope with this problem, we propose utilizing batch normalization statistics stored in the pretrained model to approximate the distribution of unobserved source data. Specifically, we fix the classifier part of the model during adaptation and only fine-tune the remaining feature encoder part so that batch normalization statistics of the features extracted by the encoder match those stored in the fixed classifier. Additionally, we also maximize the mutual information between the features and the classifier's outputs to further boost the classification performance. Experimental results with several benchmark datasets show that our method achieves competitive performance with state-of-the-art domain adaptation methods even though it does not require access to source data.

## 1 INTRODUCTION

In typical statistical machine learning algorithms, test data are assumed to stem from the same distribution as training data (Hastie et al., 2009). However, this assumption is often violated in practical situations, and the trained model results in unexpectedly poor performance (Quionero-Candela et al., 2009). This situation is called domain shift, and many researchers have intensely worked on domain adaptation (Csurka, 2017; Wilson & Cook, 2020) to overcome it. A common approach for domain adaptation is to jointly minimize a distributional discrepancy between domains in a feature space as well as the prediction error of the model (Wilson & Cook, 2020), as shown in Fig. 1(a). Deep neural networks (DNNs) are particularly popular for this joint training, and recent methods using DNNs have demonstrated excellent performance under domain shift (Wilson & Cook, 2020).

Many domain adaptation algorithms assume that they can access labeled source data as well as target data during adaptation. This assumption is essentially required to evaluate the distributional discrepancy between domains as well as the accuracy of the model's prediction. However, it can be unreasonable in some cases, for example, due to data privacy issues or too large-scale source datasets to be handled at the environment where the adaptation is conducted. To tackle this problem, a few recent studies (Kundu et al., 2020; Li et al., 2020; Liang et al., 2020) have proposed source-free domain adaptation methods in which they do not need to access the source data.

In source-free domain adaptation, the model trained with source data is given instead of source data themselves, and it is fine-tuned through adaptation with unlabeled target data so that the fine-tuned model works well in the target domain. Since it seems quite hard to evaluate the distributional discrepancy between unobservable source data and given target data, previous studies mainly focused on how to minimize the prediction error of the model with unlabeled target data, for example, by using pseudo-labeling (Liang et al., 2020) or a conditional generative model (Li et al., 2020). However, due to lack of the distributional alignment, those methods heavily depend on noisy target labels obtained through the adaptation, which can result in unstable performance.

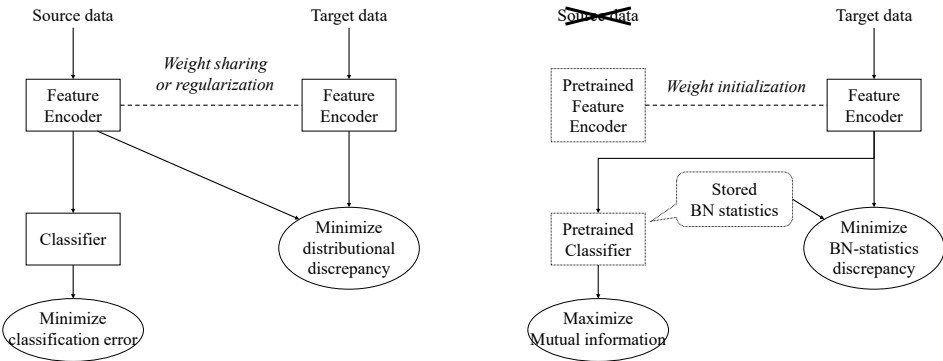

(a) General setup commonly adopted in recent typical domain adaptation methods. This visualization is inspired by (Wilson & Cook, 2020).

(b) Our setup for source-free domain adaptation.

Figure 1: Comparison between typical domain adaptation methods and our method. A rectangle with solid lines represents a trainable component, while that with dotted lines represent a fixed component during adaptation.

In this paper, we propose a novel method for source-free domain adaptation. Figure 1(b) shows our setup in comparison with that of typical domain adaptation methods shown in Fig. 1(a). In our method, we explicitly minimize the distributional discrepancy between domains by utilizing batch normalization (BN) statistics stored in the pretrained model. Since we fix the pretrained classifier during adaptation, the BN statistics stored in the classifier can be regarded as representing the distribution of source features extracted by the pretrained encoder. Based on this idea, to minimize the discrepancy, we train the target-specific encoder so that the BN statistics of the target features extracted by the encoder match with those stored in the classifier. We also adopt information maximization as in Liang et al. (2020) to further boost the classification performance of the classifier in the target domain. Our method is apparently simple but effective; indeed, we will validate its advantage through extensive experiments on several benchmark datasets.

## 2  RELATED WORK

In this section, we introduce existing works on domain adaptation that are related to ours and also present a formulation of batch normalization.

### 2.1  DOMAIN ADAPTATION

Given source and target data, the goal of domain adaptation is to obtain a good prediction model that performs well in the target domain (Csurka, 2017; Wilson & Cook, 2020). Importantly, the data distributions are significantly different between the domains, which means that we cannot simply train the model with source data to maximize the performance of the model for target data. Therefore, in addition to minimizing the prediction error using labeled source data, many domain adaptation algorithms try to align the data distributions between domains by adversarial training (Ganin et al., 2016; Tzeng et al., 2017; Deng et al., 2019; Xu et al., 2019) or explicitly minimizing a distributional-discrepancy measure (Long et al., 2015; Bousmalis et al., 2016; Long et al., 2017). This approach has empirically shown excellent performance and is also closely connected to theoretical analysis (Ben-David et al., 2010). However, since this distribution alignment requires access to source data, these methods cannot be directly applied to the source-free domain adaptation setting.

In source-free domain adaptation, we can only access target data but not source data, and the model pretrained with the source data is given instead of the source data. This challenging problem has been tackled in recent studies. Li et al. (2020) proposed joint training of the target model and the conditional GAN (Generative Adversarial Network) (Mirza & Osindero, 2014) that is to generate annotated target data. Liang et al. (2020) explicitly divided the pretrained model into two modules,

called a feature encoder and a classifier, and trained the target-specific feature encoder while fixing the classifier. To make the classifier work well with the target features, this training jointly conducts both information maximization and self-supervised pseudo-labeling with the fixed classifier. Kundu et al. (2020) adopted a similar architecture but it has three modules: a backbone model, a feature extractor, and a classifier. In the adaptation phase, only the feature extractor is tuned for the target domain by minimizing the entropy of the classifier's output. Since the methods shown above do not try to align data distributions between domains, they cannot essentially avoid confirmation bias of the model and also cannot benefit from well-exploited theories in the studies on typical domain adaptation problems (Ben-David et al., 2010).

## 2.2 BATCH NORMALIZATION

Batch normalization (BN) (Ioffe & Szegedy, 2015) has been widely used in modern architectures of deep neural networks to make their training faster as well as being stable. It normalizes each input feature within a mini-batch in a channel-wise manner so that the output has zero-mean and unit-variance. Let $B$ and $\{z_i\}_{i=1}^{B}$ denote the mini-batch size and the input features to the batch normalization, respectively. Here, we assume that the input features consist of $C$ channels as $z_i = [z_i^{(1)}, ..., z_i^{(C)}]$ and each channel contains $n_c$ features. BN first computes the means $\{\mu_c\}_{c=1}^{C}$ and variances $\{\sigma_c^2\}_{c=1}^{C}$ of the features for each channel within the mini-batch:

$$\mu_c = \frac{1}{n_c B} \sum_i^B \sum_j^{n_c} z_i^{(c)}[j], \ \sigma_c^2 = \frac{1}{n_c B} \sum_i^B \sum_j^{n_c} (z_i^{(c)}[j] - \mu_c)^2, \tag{1}$$

where $z_i^{(c)}[j]$ is the $j$-th feature in $z_i^{(c)}$. Then, it normalizes the input features by using the computed BN statistics:

$$\tilde{z}_i^{(c)} = \frac{z_i^{(c)} - \mu_c}{\sqrt{\sigma_c^2 + \epsilon}}, \tag{2}$$

where $\epsilon$ is a small positive constant for numerical stability. In the inference phase, BN cannot always compute those statistics, because the input data do not necessarily compose a mini-batch. Instead, BN stores the exponentially weighted averages of the BN statistics in the training phase and uses them in the inference phase to compute $\tilde{z}$ in Eq. (2) (Ioffe & Szegedy, 2015).

Since BN renormalizes features to have zero-mean and unit-variance, several methods (Li et al., 2018; Chang et al., 2019; Wang et al., 2019) adopted domain-specific BN to explicitly align both the distribution of source features and that of target features into a common distribution. Since the domain-specific BN methods are jointly trained during adaptation, we cannot use these methods in the source-free setting.

## 3 PROPOSED METHOD

Figure 2 shows an overview of our method. We assume that the model pretrained with source data is given, and it conducts BN at least once somewhere inside the model. Before conducting domain adaptation, we divide the model in two sub-models: a feature encoder and a classifier, so that BN comes at the very beginning of the classifier. Then, for domain adaptation, we fine-tune the encoder with unlabeled target data with the classifier fixed. After adaptation, we use the fine-tuned encoder and the fixed classifier to predict the class of test data in the target domain.

To make the fixed classifier work well in the target domain after domain adaptation, we aim to obtain a fine-tuned encoder that satisfies the following two properties:

- The distribution of target features extracted by the fine-tuned encoder is well aligned to that of source features extracted by the pretrained encoder.
- The features extracted by the fine-tuned encoder are sufficiently discriminative for the fixed classifier to accurately predict the class of input target data.

To this end, we jointly minimize both the BN-statistics matching loss and information maximization loss to fine-tune the encoder. In the former loss, we approximate the distribution of unobservable

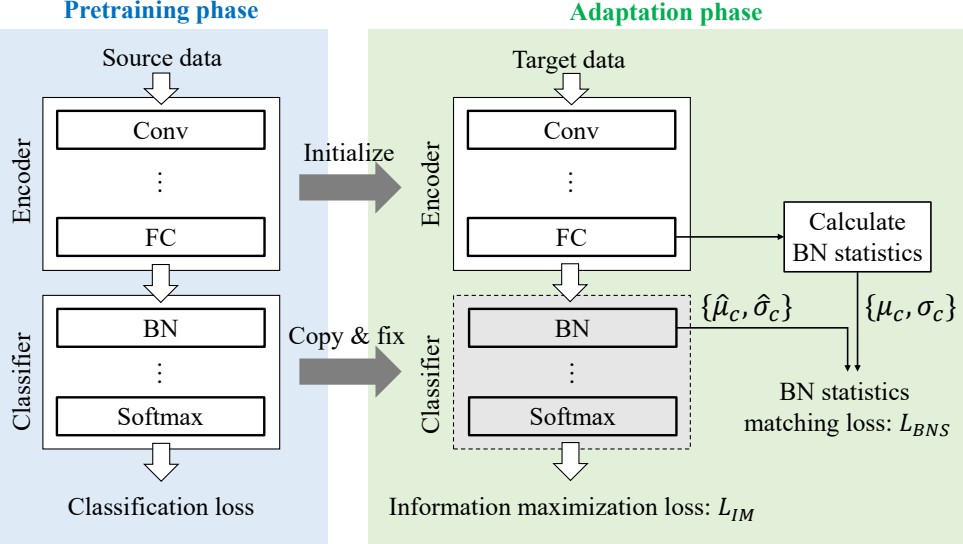

Figure 2: An overview of the proposed method.

source features by using the BN statistics stored in the first BN layer of the classifier, and the loss explicitly evaluates the discrepancy between source and target feature distributions based on those statistics. Therefore, minimizing this loss leads to satisfying the first property shown above. On the other hand, the latter loss is to make the predictions by the fixed classifier certain for every target sample as well as diverse within all target data, and minimizing this loss leads to fulfilling the second property. Below, we describe the details of these losses.

## 3.1 DISTRIBUTION ALIGNMENT BY MATCHING BATCH NORMALIZATION STATISTICS

Since the whole model is pretrained with source data and we fix the classifier while finetuning the encoder, the BN statistics stored in the first BN in the classifier can be seen as the statistics of the source features extracted by the pretrained encoder. We approximate the source-feature distribution by using these statistics. Specifically, we simply use a Gaussian distribution for each channel denoted by $\mathcal{N}(\hat{\mu}_c, \hat{\sigma}_c^2)$ where $\hat{\mu}_c$ and $\hat{\sigma}_c^2$ are the mean and variance of the Gaussian distribution which are the stored BN statistics corresponding to the $c$-th channel.

To match the feature distributions between domains, we define the BN-statistics matching loss, which evaluates the averaged Kullback-Leibler (KL) divergence from the target-feature distribution to the approximated source-feature distribution:

$$
\begin{aligned}
L_{\text{BNM}}(\{x_i\}_{i=1}^B, \theta) &= \frac{1}{C} \sum_{c=1}^C \text{KL}\left(\mathcal{N}(\hat{\mu}_c, \hat{\sigma}_c^2) \| \mathcal{N}(\mu_c, \sigma_c^2)\right) \\
&= \frac{1}{2C} \sum_{c=1}^C \left( \log \frac{\sigma_c^2}{\hat{\sigma}_c^2} + \frac{\hat{\sigma}_c^2 + (\hat{\mu}_c - \mu_c)^2}{\sigma_c^2} - 1 \right),
\end{aligned}
\tag{3}
$$

where $\{x_i\}_{i=1}^B$ is a mini-batch from the target data, $\theta$ is a set of trainable parameters of the encoder, and $\mu_c$ and $\sigma_c$ are the BN statistics of the $c$-th channel computed from the target mini-batch. Note that, since $\mu_c$ and $\sigma_c$ are calculated from the features extracted by the encoder, they depend on $\theta$. Here, we also approximate the target-feature distribution with another Gaussian distribution so that the KL divergence can be efficiently computed in a parametric manner. By minimizing this loss, we can explicitly reduce the discrepancy between the distribution of unobservable source features and that of target features.

In Eq. (3), we chose the KL divergence to measure the distributional discrepancy between domains. There are two reasons for this choice. First, the KL divergence between two Gaussian distributions

is easy to compute with the BN statistics as shown in Eq. (3). Moreover, since these statistics are naturally computed in the BN layer, calculating this divergence only requires tiny calculation costs. Secondly, it would be a theoretically-inspired design from the perspective of risk minimization in the target domain. When we consider a binary classification task, the expected risk of any hypothesis $h$ in the target domain can be upper-bounded under some mild assumptions as the following inequality (Ben-David et al., 2010):

$$r_{\mathrm{T}}(h) \leq r_{\mathrm{S}}(h) + d_1(p_{\mathrm{S}}, p_{\mathrm{T}}) + \beta, \tag{4}$$

where $r_{\mathrm{S}}(h)$ and $r_{\mathrm{T}}(h)$ denote the expected risk of $h$ under the source-data distribution $p_{\mathrm{S}}$ and target-data distribution $p_{\mathrm{T}}$, respectively, $d_1(p, q)$ represents the total variation distance between $p$ and $q$, and $\beta$ is a constant value that is expected to be sufficiently small. This inequality roughly gives a theoretical justification to recent domain adaptation algorithms, that is, joint minimization of both the distributional discrepancy between domains (corresponding to the second term of the bound in Eq. (4)) and the prediction error of the model (corresponding to the first term of the bound in Eq. (4)). Here, the total variation distance can be related to the KL divergence by Pinsker's inequality (Csiszar & Körner, 2011):

$$d_1(p, q) \leq \sqrt{\frac{1}{2}\mathrm{KL}(p||q)}. \tag{5}$$

Consequently, we can guarantee that minimizing the KL divergence between domains minimizes the bound of the target risk.

## 3.2 DOMAIN ADAPTATION WITHOUT ACCESS TO THE SOURCE DATA

Only aligning the marginal feature-distributions between domains does not guarantee that the fixed classifier works well in the target domain, because the features extracted by the encoder are not necessarily discriminative. If the features are sufficiently discriminative for the classifier, we can expect that the output of the classifier is almost always a one-hot vector but is diverse within the target data. Therefore, following the approach presented in Liang et al. (2020), we also adopt the information maximization loss to make the classifier work accurately.

$$L_{\mathrm{IM}}(\{x_i\}_{i=1}^{B}, \theta) = -H\left(\frac{1}{B}\sum_{i}^{B} f_\theta(x_i)\right) + \frac{1}{B}\sum_{i}^{B} H\left(f_\theta(x_i)\right), \tag{6}$$

where $H(p(y)) = -\sum_{y'} p(y') \log p(y')$ is the entropy function, and $f_\theta(x)$ denotes the output of the classifier. The first term in the right-hand side of Eq. (6) is the negative entropy of the averaged output of the classifier, and minimizing it leads to large diversity of the output within the mini-batch. The second term is the averaged entropy of the classifier's output, and minimizing it makes the outputs close to one-hot vectors. Therefore, the features extracted by the target encoder are induced to be discriminative by minimizing the information maximization loss.

Finally, our source-free domain adaptation method is formulated as joint minimization of both the BN-statistics matching loss in Eq. (3) and the information maximization loss in Eq. (6):

$$\min_\theta \mathbb{E}_{\{x_i\}_{i=1}^{B} \sim \mathcal{D}_t} \left[ L_{\mathrm{IM}}(\{x_i\}_{i=1}^{B}, \theta) + \lambda L_{\mathrm{BNM}}(\{x_i\}_{i=1}^{B}, \theta) \right], \tag{7}$$

where $\mathcal{D}_t$ is the target dataset from which the mini-batch is sampled, and a hyper-parameter $\lambda$ controls the balance between the two terms. Note that this optimization can be conducted without the source data, which means that we do not need to access to the source data during adaptation.

## 4 EXPERIMENTS

We conducted experiments with several datasets that are commonly used in existing works on domain adaptation. Specifically, we used digit recognition datasets (MNIST (LeCun et al., 1998), USPS (LeCun et al., 1990), and SVHN (Netzer et al., 2011)) and an object recognition dataset (Office-31 dataset (Saenko et al., 2010)). In the experiment, we first pretrained the model with the source training data. Following the setup in Liang et al. (2020), we used standard cross-entropy loss with label smoothing for this pretraining. Then, we apply our source-free domain adaptation

Table 1: Experimental results with Office-31 dataset. The bold number represents the highest test accuracy among the source-free domain adaptation methods, and the underline represents the second highest one.

| Method | A→D | A→W | D→A | D→W | W→A | W→D |
|---|---|---|---|---|---|---|
| SHOT (Liang et al., 2020) | **94.0** | 90.1 | 74.7 | 98.4 | 74.3 | 99.9 |
| Model adaptation (Li et al., 2020) | 92.7±0.4 | **93.7**±0.2 | 75.3±0.5 | 98.5±0.1 | **77.8**±0.1 | 99.8±0.2 |
| Our method | 89.0±0.2 | 91.7±1.0 | **78.5**±0.2 | **98.9**±0.1 | 76.6±0.7 | **100.0**±0.0 |
| ADDA (Tzeng et al., 2017) | 77.8±0.3 | 86.2±0.5 | 69.5±0.4 | 96.2±0.3 | 68.9±0.5 | 98.4±0.3 |
| rRevGrad+CAT (Deng et al., 2019) | 90.8±1.8 | 94.4±0.1 | 72.2±0.6 | 98.0±0.1 | 70.2±0.1 | 100.0±0.0 |
| $d$-SNE (Xu et al., 2019) | 94.7±0.4 | 96.6±0.1 | 75.5±0.4 | 99.1±0.2 | 74.2±0.2 | 100.0±0.0 |

method to fine-tune the pretrained model with the target training data. We used Adam optimizer for both pretraining and adaptation. The number of iterations in the optimization was set to 30,000, and the batch size was set to 64. The hyper-parameter $\lambda$ in Eq. (7) is set to 10 in all experiments except for those in section 4.3. The performance of the domain adaptation is evaluated by test accuracy of the fine-tuned model on the target test data. We report the averaged accuracy as well as the standard deviation over five runs with random initialization of the model at the pretraining phase.

We compared the performance of our method with those of the state-of-the-art methods for source-free domain adaptation (Li et al., 2020; Liang et al., 2020), which are most related to our work. We did not include the work by Kundu et al. (2020) in this comparison, because it is designed for more difficult setting, called universal domain adaptation. For reference, we also show the performance of the recent methods for typical domain adaptation (Tzeng et al., 2017; Deng et al., 2019; Xu et al., 2019), though they require access to the source data during adaptation.

## 4.1 OBJECT RECOGNITION DATASET

The Office-31 dataset comprises three domains: Amazon (A), DSLR (D), and Webcam (W). We examined all possible combinations for the adaptation, which results in six scenarios. Following the setup of Ganin et al. (2016); Liang et al. (2020), we used ResNet-50 pretrained with the ImageNet classification dataset as a backbone model. We removed the original FC layer from the pretrained ResNet-50 and added a bottleneck FC layer (256 units) and a classification FC layer (31 units). A BN layer is put before and after the bottleneck layer, and we used the last one to calculate our BN-statistics matching loss. Note that the backbone part is fixed in our experiments.

Table 1 shows the test accuracy of the adapted models at the target data. The results shown above the double line are those of the source-free domain adaptation methods, while the remaining ones are those of the other typical domain adaptation methods. Our method achieved the best accuracy at three out of six scenarios, and, surprisingly, its performance reached or exceeded the performance of the state-of-the-art typical domain adaptation methods in those cases. Moreover, our method also shows competitive performance in the other scenarios except for A → D. Since SHOT (Liang et al., 2020) also adopts the information maximization loss, these results indicate that our BN-statistics matching loss substantially improves the performance of the adaptation by successfully reducing the distributional discrepancy between domains. The model adaptation (Li et al., 2020) also works well through the all scenarios. However, considering that it requires training of a conditional GAN while adaptation, our method is quite appealing due to simplicity of its training procedure as well as its high performance.

## 4.2 DIGIT RECOGNITION DATASETS

We examined USPS ↔ MNIST and SVHN → MNIST scenarios. Following the previous studies (Long et al., 2018; Liang et al., 2020), we used the classical LeNet-5 network for the former scenario,

Table 2: Experimental results with digit recognition datasets. The bold number represents the highest test accuracy among the source-free domain adaptation methods, and the underline represents the second highest one.

| Method | USPS→MNIST | MNIST→USPS | SVHN→MNIST |
|---|---|---|---|
| SHOT (Liang et al., 2020) | 98.4±0.6 | **98.0**±0.2 | 98.9±0.0 |
| Model adaptation (Li et al., 2020) | **99.3**±0.1 | 97.3±0.2 | **99.4**±0.1 |
| Our method | 99.1±0.0 | 97.7±0.2 | 99.1±0.0 |
| ADDA (Tzeng et al., 2017) | 90.1±0.8 | 89.4±0.2 | 76.0±1.8 |
| rRevGrad+CAT (Deng et al., 2019) | 96.0±0.9 | 94.0±0.7 | 98.8±0.0 |
| $d$-SNE (Xu et al., 2019) | 98.5±0.4 | 99.0±0.1 | 96.5±0.2 |

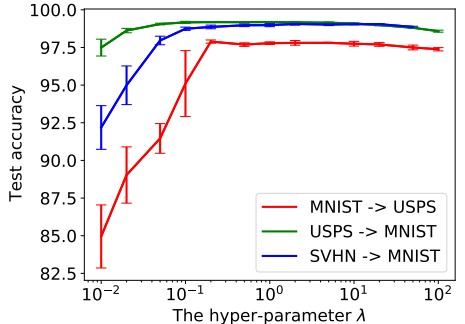
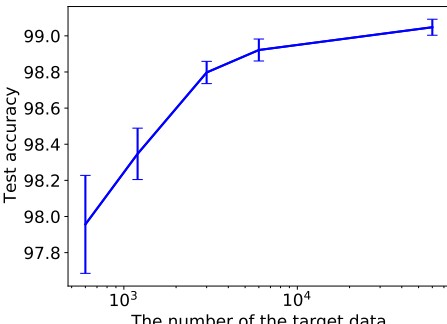

Figure 3: Sensitivity of the performance to the hyper-parameter $\lambda$.

Figure 4: Sensitivity of the performance to the dataset size.

while a variant of LeNet, called DTN, is used for the latter one. For both models, we used the last BN layer in the model to calculate the BN statistics matching loss in our method.

Table 2 shows the experimental results with the digit recognition datasets. Although our method did not achieve the best performance among the source-free methods, it stably achieved the second highest accuracy in all scenarios. Similarly in the results with Office-31 dataset, our method exceeds the performance of the typical domain adaptation methods in two scenarios, namely USPS→MNIST and SVHN→MNIST.

### 4.3 PERFORMANCE SENSITIVITY TO THE HYPER-PARAMETER AND DATASET SIZE

We investigated the performance sensitivity of our method to the hyper-parameter setting and that to the size of the target dataset. The experimental settings are same with those in the previous experiment unless otherwise noted.

Our method introduces single hyper-parameter, which is $\lambda$ in Eq. (7). We first investigated the performance sensitivity to the value of $\lambda$. Since we can only access the unlabeled target data during adaptation, it is essentially hard to appropriately tune the hyper parameter. Therefore, high stability of the performance under a suboptimal setting of the hyper-parameter is required in the source-free domain adaptation. In the experiment, we varied the value of $\lambda$ from $0.01$ to $100$ and used it in our method to conduct the adaptation with the digit recognition datasets. Figure 3 shows how the test accuracy of the adapted model changes according to the value of $\lambda$. In all adaptation scenarios, the performance of our method is quite stable against the change of the value of $\lambda$. It keeps almost same within the wide range of the value of $\lambda$, specifically $0.2 \leq \lambda \leq 50$.

We also investigated the performance of our method in case of small-scale target data. This investigation is crucial, because, considering the motivation of domain adaptation, we cannot always expect sufficiently large amount of the target data. Since MNIST is the largest target dataset used in our experiments, we conducted this investigation with SVHN→MNIST adaptation. To make the small-scale target data, we randomly selected a subset of the original target training data while

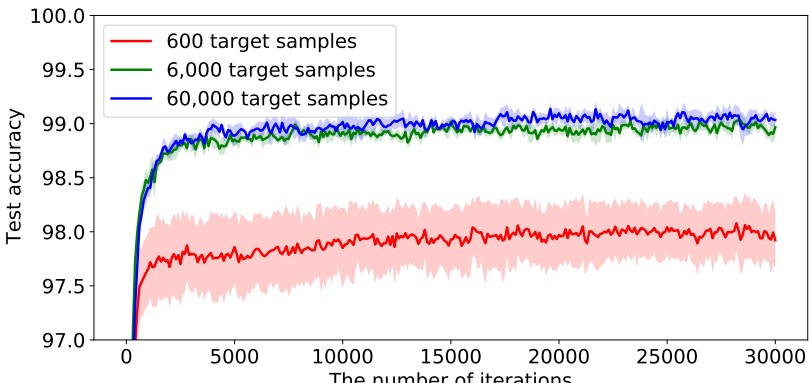

Figure 5: Test accuracy curves during adaptation in our method.

keeping the class prior same with that in the original dataset. Figure 4 shows how the test accuracy after the adaptation changes according to the size of the target dataset. As decreasing the number of the target data, the performance of our method becomes deteriorated to some extent. However, even when there are only 600 samples in the target dataset, our method still achieved $98.0\%$, which is comparable performance with those of the typical domain adaptation methods using full target dataset as well as source dataset.

Figure 5 shows how the test accuracy by the model changes during adaptation. The accuracy is stably and monotonically improved even when the number of the target data is small. It means that our method can effectively avoid overfitting to the small-scale target dataset.

## 5 CONCLUSION

We proposed a novel domain adaptation method for source-free setting. To match the distributions between unobservable source data and given target data, we utilize the BN statistics stored in the pretrained model to explicitly estimate and minimize the distributional discrepancy between domains. This approach is quite efficient in terms of the computational cost and can be justified from the perspective of risk minimization in the target domain. Experimental results with several benchmark datasets have shown that our method performs well even though it does not require the access to the source data. Moreover, its performance was empirically quite stable against suboptimal hyper-parameter setting or limited size of the target dataset. In conclusion, we argue that our method is quite promising to tackle many real-world problems that are hard to solve with existing domain adaptation methods.

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
