# OpenReview forum: "Source-free Domain Adaptation via Distributional Alignment by Matching Batch Normalization Statistics"
_ICLR.cc/2021/Conference — Reject_

### Official Review · AnonReviewer2 · 2020-10-28
**Domain adaptation with meta information when source data is not available**

**Rating:** 6
**Confidence:** 3

**Review:**

### Summary
This paper proposes a domain adaptation technique when source data is not available. The exponentially weighted average of BN statistics from source training along with the trained model is utilized to align source and target distributions. Source model is divided into feature encoder and classifier components based on the presence of the last BN layer. BN statistics matching loss minimizes the distribution discrepancy between source and target, whereas information maximization loss enforces the classifier to be sufficiently discriminative. Experiments on several benchmark datasets showed competitive performance with state-of-the-art domain adaptation methods.

### Strong Points
 1. Sound modelling with thorough experiments on benchmark datasets along with small-scale datasets
 2. Application of joint optimization ensures discrimitiveness between classes at the same time ensuring domain matching.

### Weak Points
 1. As classifier part of the model is fixed I am curious how the model handles label shift (i.e. when labels distribution differs between source and target)
 2. How does the technique performs when source model has multiple BN layers and till the first BN layer is considered as encoder will be interesting. Does choosing the last BN always perform better than first BN?

---

> ### Author Response · Authors · 2020-11-18
> **Response to Reviewer 2**
>
> Thank you for your time reading our paper and providing valuable comments. Please find our response shown below.
>
>
> Weak point 1: label shift
>
> Our method can work to some extent even when a label shift occurs. For example, in experiments with digit recognition datasets, our method works well in the SVHN->MNIST adaptation, though the number of samples for each class is balanced in MNIST but is somewhat imbalanced in SVHN (since samples in SVHN come from house numbers, the number of “0” and “1” is relatively large). However, if the label distributions are substantially different between domains, our method would fail, because it basically aims to match marginal distributions between domains.
>
>
> Weak point 2: multiple BN layers
>
> In experiments we conducted so far, choosing the last BN always performs best. We conjecture that choosing early ones degrades the performance by the following two reasons. First, it reduces the flexibility of the feature encoder, which would result in insufficient distributional alignment between domains. Second, it makes regularization by BN-statistics matching looser, because the number of channels (C in Eq. (3)) tends to be small in an early stage in popular network architectures.
>
>
> Weak point 3: requirements of BN statistics
>
> A BN layer implemented in almost all deep learning frameworks (e.g., pyTorch and TesnsorFlow) stores BN statistics by default to use them in the inference phase. Therefore, there is no special requirement for our method except for the usage of BN in the model as stated in the beginning of Section 3. Since BN layers are widely used in modern neural network architectures, our method is applicable to a wide range of neural networks, which is one of the advantages of our method.

---

### Official Review · AnonReviewer3 · 2020-10-28
**I appreciated the first new part,  but I felt disappointed of the second part previously proposed by others, the experiment results and the inadequate ablation studies.**

**Rating:** 4
**Confidence:** 4

**Review:**

In the work, the authors focus on tackling the problem of source free domain adaptation. The proposed method mainly has two parts, in which the second is nearly the same as the SHOT-IM as in Liang et al., 2020 [1], while the first part aims at coping with this problem from a new perspective to align the distribution of target features extracted by the fine-tuned encoder to that of source features extracted by the pre-trained encoder. To achieve this, they utilize batch normalization statistics stored in the pre-trained model to approximate the distribution of unobserved source data.

Pros:
1.	Tackling the problem of source free domain adaptation from a new perspective of aligning the distribution of target features extracted by the fine-tuned encoder to that of source features extracted by the pre-trained encoder, specifically, the BN statistic. They also provide a roughly promising theoretical analysis. To me, the first part is new, elegant, and interesting.
2.	This paper is well-written and crystal-clear, making it enjoyable to read.

Before reading the second part of information maximization loss proposed by Liang et al. 2020 [1], which also tackles the source free domain adaptation, I really want to accept this paper. However, after reading the remaining part, the drawbacks of this paper are too obvious to be ignored. My major concerns can be concluded as follows:

1.	The second part of information maximization loss is nearly the same as that of SHOT-IM as in Liang et al., 2020 [1], resulting in a limited novelty of the whole paper. To me, the second part should at least provide some insights or some different perspectives to make the technique novelty enough for a top-tier conference.
2.	Meanwhile, the experiments section also makes me a little disappointed. Neither state-of-the-art results nor comprehensive ablation studies are seen. First, in most domain adaptation tasks, the performance of the proposed method is obviously lower than SHOT Liang et al., 2020 [1], and Model adaptation (Li et al., 2020) [2]. For a new paper that meets the bottom line of a top-tier conference, at least little improvement should be seen in most tasks.
3.	Further, since the proposed method consists of two parts, some basic ablation studies should be conducted to verify the effectiveness of both parts. Without taking apart the whole method, we will never know how much improvement each part contributes.
4.	 Another concern is that the performance sensitivity experiment is only conducted on a simple task SVHN→MNIST, which is too weak to draw a conclusion that the proposed method keeps almost the same within the wide range of the value of λ, specifically 0.1 ≤ λ ≤ 50.


[1] Jian Liang, Dapeng Hu, and Jiashi Feng. Do we really need to access the source data? source hypothesis transfer for unsupervised domain adaptation. In International Conference on Machine Learning, 2020.

[2] Rui Li, Qianfen Jiao, Wenming Cao, Hau-San Wong, and Si Wu. Model adaptation: Unsupervised domain adaptation without source data. In Proceedings of the IEEE/CVF Conference on Computer Vision and Pattern Recognition, pp. 9641–9650, 2020.

---

> ### Author Response · Authors · 2020-11-18
> **Response to Reviewer 3**
>
> Thank you for your time reading our paper and providing valuable comments. Please find our response shown below.
>
>
> Con 1: novelty of this work
>
> As you mentioned, the idea to use the information maximization loss itself is not novel. Ineed, it has been widely used in unsupervised learning methods, for example, in Hu et al. [2017], and we used it as a baseline in this work. Our novelty is mainly on introducing the BN-statistics matching loss. By adopting it, we can explicitly bridge source-free domain adaptation and standard domain adaptation in theory, and it substantially improves the performance in the experiments. Additionally, since our method does not require any special module or training for pretrained models except for well-known BN, we can utilize many publicly-available pretrained models for source-free domain adaptation, which is highly useful in practice. On the other hand, Liang et al. [2020] requires weight normalization at the last layer of the pretrained model, which is not so popular compared with BN.
>
> [Hu et al., 2017] “Learning Discrete Representations via Information Maximizing Self-Augmented Training,” ICML 2017.
>
> Con 2: the performance of our method
>
> When we compare our method with SHOT [Liang et al., 2020], our method outperforms it in 7 out of 9 scenarios. Therefore, we argue that our method achieves better performance than SHOT.
> When we compare our method with Model Adaptation [Li et al., 2020], our method is superior in 4 out of 9 scenarios. Therefore, the performance of our method would be on par with Model Adaptation. However, as stated at the end of Section 4.1, considering that Model Adaptation requires training of a conditional GAN during adaptation, our method is quite appealing due to simplicity of its training procedure.
>
> Con 3: ablation study
>
> As you pointed out, although we have shown the performance with various settings of lambda, we did not show the performance with only using the BN-statistics matching loss or information maximization loss. However, we conjecture that only using one of the two losses results in quite poor performance by the following reasons:
> - When we use only the BN-statistics matching loss
> As stated at the beginning of Section 3.2, since minimizing the BN-statistics matching loss only matches marginal feature distributions between domains, the extracted features are not necessarily discriminative, which results in poor performance of the fixed classifier. For example, in case of the SVHN->MNIST adaptation, the test accuracy was only 20.7.
> - When we use only the information maximization loss
> The proposed method works to some extent, but the performance is not so good. For example, in case of the SVHN->MNIST adaptation, the test accuracy was 87.4. This setting is quite similar to that reported in Liang et al. [2020] as SHOT-IM.
>
> Con 4: performance sensitivity to the hyper-parameter
>
> We conducted additional experiments to show the performance sensitivity to lambda in other datasets. Our method also showed similar behaviors to those reported in the paper, and we succeeded in validating high robustness of our method against suboptimal settings of lambda. We included this result in the revised manuscript.

---

### Official Review · AnonReviewer5 · 2020-11-07
**Cool idea, some questions about comparison with Liang et al 2020**

**Rating:** 6
**Confidence:** 3

**Review:**

After author response: I've read the response and other reviews and keep my score (weak accept). R3 says that "The second part of information maximization loss is nearly the same as that of SHOT-IM", but I don't see it as a problem. The first part seems interesting and novel, and to get strong results they need to combine it with an existing technique, which seems typical. I agree with the authors and disagree with R3: that their "method outperforms it (SHOT) in 7 out of 9 scenarios." So the positive is that it's a nice idea, simple method, and performs quite well.

However, I agree with R3 that there should be more extensive ablations to understand the effect of each part, and the sensitivity analysis of lambda should be done on more datasets. Additionally, I'd like to see more detailed comparisons to related work like https://arxiv.org/abs/2006.10963 that uses batchnorm for domain adaptation. The results aren't stellar, so without a good conceptual explanation, or empirical investigation, I don't see this as a must accept.

#########################################################################

Summary:

This paper tackles the problem of unsupervised domain adaptation, where there may be privacy constraints on the source, so we have access to a source model but not the source data. Prior work (Liang et al) essentially adapts the model by entropy minimization / self-training on the unlabeled data. This paper proposes in addition aligning the batchnorm statistics between the source and target data (inspired by domain adversarial training). They see improvements in some digits and office adaptation tasks.

#########################################################################

Reasons for score:

Aligning batch-norm statistics is a simple and nice idea, and this direction seems promising. The method even performs comparably to regular unsupervised domain adaptation methods (no privacy), that use domain adversarial training and are notoriously difficult to tune so this approach could simplify tuning in addition to preserving privacy. They show small improvements on 7/9 tasks compared to Liang et al 2020. It’s an interesting piece of work!

#########################################################################

Pros:

- I like the idea of aligning batch norm statistics when we do not have access to the source data and cannot do domain adversarial training.

- Results seem quite promising, especially compared to the closest related work (Liang et al). Some other methods do better, but they seem substantially more complicated.

#########################################################################

Cons:

- It’s not easy for me to tell if there are other differences relative to Liang et al 2020 besides the addition of the L_{BNM} loss. Are the regularization, augmentation, number of epochs, etc trained the same? I noticed that the numbers in table 1 and 2 are taken directly from Liang et al. Given that the numbers aren’t clearly better (7/9 cases, and by small amounts), I’d like a more clear ablation to see that the improvement is indeed coming from L_{BNM}, and not e.g. because of a different data augmentation strategy.

- Relatedly to the previous point, is Liang et al the same as your method in the case when lambda = 0? However, it seems from Figure 3, that the accuracy when you set lambda -> 0 is very low (around < 93%). But Liang et al get 98.9% on SVHN -> MNIST. Can you explain this discrepancy?

- I’d compare to this simpler method: where you naively normalize the representations in the target so that the statistics match the source. That is, if mu_c, sigma_c is the target statistic for channel c, and \hat{mu}_c, \hat{sigma}_c is the source statistic for channel c, then just normalize the feature, so if x_c is the value of channel c for an example, then normalize it to get [(x_c - mu_c) / sigma_c] * \hat{sigma}_c + \hat{mu}_c. Effectively, this is explicitly / directly aligning the source and target batchnorm statistics. After doing this explicit alignment, you can then just train on the loss L_{IM} to further refine the model. This would shed insight into whether you need to jointly optimize L_{IM} and L_{BNM}, or can just do the explicit alignment of batchnorm statistics and then optimize L_{IM}.

#########################################################################

Questions and things to improve:

- This didn’t affect my score, but the claims about theoretical connections need to be toned down substantially. It’s fine for a method to be a heuristic that is loosely inspired by theory, but the paper just needs to be upfront about this.

For example, page 1 says “, the performance of those methods is not guaranteed by theories exploited in typical domain adaptation studies”, however a number of recent works have explained why self-training based methods do work, e.g. Kumar et al 2020, Chen et al 2020. Page 2 says domain adversarial training “is also validated in theory”. This is too strong, domain adversarial training is only inspired by theory and not validated by it. The theory in Ben-David et al and follow up works assume that there exists a classifier that does well on both the source and the target - and even if this is true in the input space, it may not be true in the learned representation space obtained from domain adversarial training. See Zhao et al 2019. In the conclusion, the paper says the method is “theoretically validated”, but the paper does not give any theoretical explanation for why the method works. This can be changed to “theoretically inspired” if you like. Similarly the use of “theoretically justified” should be toned down in section 3.1 (e.g. top of page 5)

#########################################################################

A. Kumar, T. Ma, P. Liang. Understanding Self-Training for Gradual Domain Adaptation. ICML 2020.

Y. Chen, C. Wei, A. Kumar, T. Ma. Self-training Avoids Using Spurious Features Under Domain Shift. NeurIPS 2020.

H. Zhao, R. T. des Combes, K. Zhang, and G. J. Gordon. On learning invariant representation for domain adaptation. ICML 2019.

---

> ### Author Response · Authors · 2020-11-18
> **Response to Reviewer 5**
>
> Thank you for your time reading our paper and providing valuable comments. Please find our response shown below.
>
> Con 1: experimental settings
>
> We basically followed the same setup as in Liang et al. [2020], including model architectures, the data augmentation strategy, and how to pretrain the model with source data. However, several points are different as shown below:
> - We did not use weight decay, because our BN-statistics matching loss has a similar regularization effect. It prevents features from taking an extremely large value by matching the variance between domains.
> - As described at the beginning of Section 4, we used the Adam optimizer for training, while Liang et al. [2020] used SGD with momentum and an exponentially-decreasing learning rate. Since their method conducts pseudo-labeling during training, they need to carefully tune the learning rate to prevent incorrect pseudo labels. On the other hand, our method does not require such careful tuning, which makes our method easy-to-use.
> - As described at the beginning of Section 4, the number of iterations in our experiments is set to 30,000 for all experiments, while Liang et al. [2020] set it differently for each experiment. We did not carefully tune this, and, as shown in Fig. 5, it would be possible to reduce it while keeping the performance. Our method has shown quite stable and fast convergence.
>
> How our BN-statistics matching loss contributes to the performance has shown in Fig. 3 for the SVHN->MNIST adaptation scenario. We added more results with other datasets in the revised manuscript.
>
>
> Con 2: difference from Liang et al. [2020]
>
> Our method with lambda = 0 is not same as SHOT [Liang et al., 2020]. The biggest difference is that SHOT uses self-supervised pseudo-labeling in addition to information maximization for further improvement of the classification performance. In Liang et al. [2020], the performance of SHOT without pseudo-labeling, called SHOT-IM, has also been reported and is substantially worse than SHOT. For example, in case of the SVHN->MNIST adaptation scenario, the performance is degraded from 98.9 (SHOT) to 89.6 (SHOT-IM).
>
>
> Con 3: simpler baseline
>
> As far as we understand, it should result in the standard BN. Since “[(x_c - mu_c) / sigma_c] * \hat{sigma}_c + \hat{mu}_c” are not normalized features, we need an additional normalization process to them, which leads to the same result as features normalized with the target BN statistics.
>
> About questions and things to improve:
>
> We really appreciate your advice. We have changed our wordings to more appropriate ones in the revised manuscript.

---

> > ### Comment · AnonReviewer5 · 2020-11-23
> > **Thanks for the clarifications, simpler baseline is not regular batchnorm**
> >
> > Thanks for clarifying the differences with Liang et al. Since there are many differences even between this method and SHOT-IM, it would be good to isolate where the gains are coming from. However, Figure 3 does a good job at this for the digits datasets, can you plot something similar for OFFICE, to show that the batchnorm loss helps?
> >
> > I don't think the simpler baseline I suggested is regular batchnorm. Batchnorm uses the running average of the training set (the source dataset) at inference time. It seems like that is the case in your implementation of Batchnorm as well, from Section 2.2. My suggestion is a simple, explicit way to align the source and target representations in the batchnorm layer. It looks like this has been used for domain adaptation, for example, see https://arxiv.org/abs/2006.10963. As in Figure 4, the max benefit occurs when you use the whole target set when computing these statistics, which is what I wrote in the review above. I think this is useful to compare with because it's 1) an existing method for domain adaptation by normalizing batchnorm statistics, and 2) it's much simpler and very explicitly aligns the statistics.
> >
> > Another question is why your alignment method needs to be specifically applied to a batchnorm layer. Can't you do this at any layer?

---

> > > ### Author Response · Authors · 2020-11-24
> > > **Thank you for your response and clarification.**
> > >
> > > > can you plot something similar for OFFICE, to show that the batchnorm loss helps?
> > >
> > > Thank you for your suggestion. We are conducting additional experiments for this plot. However, since we prioritized another experiment on "simpler baseline," those results might not be obtained in time. Once obtained, we will update the manuscript.
> > >
> > > > I don't think the simpler baseline I suggested is regular batchnorm.
> > >
> > > We really appreciate your clarification. We misunderstood your suggestion.
> > >
> > > We tried this in SVHN->MNIST scenario, but its performance was 75.0%, which is worse than only using information maximization loss (87.4% as stated in the response to reviewer 3). We conjecture that, since aligning BN statistics only matches marginal feature distributions between domains, the extracted features are not necessarily discriminative, which leads to such worse performance.
> > >
> > > > Another question is why your alignment method needs to be specifically applied to a batchnorm layer. Can't you do this at any layer?
> > >
> > > Yes, we can do a similar thing at any layer, but, in that case, we need to make additional effort to store batch statistics at the target layer in a pretraining phase, which limits applicability of our method.
> > >
> > > A BN layer implemented in almost all deep learning frameworks (e.g., pyTorch and TesnsorFlow) stores BN statistics by default to use them in the inference phase. Therefore, there is no special requirement for our method except for the usage of BN in the model as stated in the beginning of Section 3. Since BN layers are widely used in modern neural network architectures, our method is applicable to a wide range of neural networks, which is one of the advantages of our method.

---

### Decision · Program_Chairs · 2021-01-07
**Final Decision**

**Decision:**

Reject

**Comment:**

This submission develops a novel technique for domain adaptation for the setup where only a trained model (but no data) from the source task is available. The authors propose to fine-tune the feature encoder using batch norm statistics of the features extracted. Additionally their criterion also promotes increasing the the mututal information between features and target classification. The developed method is experimentally evaluated on several benchmarks.

Pros:
- The problem considered is of practical relevance and general interest in ICLR community
- The proposed methodology is well motivated and shows good performance

Cons:
- There is no thorough formal analysis of when the method would work and not work; not even on an intuitive level (state conditions under which the proposed method should be expected to work better/worse than other state of the art optimization criteria for the same setup
- Alternatively to a sound theoretical analysis, the authors should provide a more extensive set of ablation experiments (this was mentioned by several reviewers)

In the current format, it remains unclear, how the research community would benefit from the study presented.